# Strategies to Reduce Salt Content: PDO and PGI Meat Products Case

**DOI:** 10.3390/foods13172681

**Published:** 2024-08-25

**Authors:** Maria João Fraqueza, Cristina Mateus Alfaia, Sandra Sofia Rodrigues, Alfredo Teixeira

**Affiliations:** 1CIISA—Centro de Investigação Interdisciplinar em Sanidade Animal, Faculdade de Medicina Veterinária, Universidade de Lisboa, Av. da Universidade Técnica, 1300-477 Lisboa, Portugal; cpmateus@fmv.ulisboa.pt; 2Associate Laboratory for Animal and Veterinary Sciences (AL4AnimalS), 1300-477 Lisboa, Portugal; 3CIMO—Centro de Investigação de Montanha, Instituto Politécnico de Bragança, Campus de Santa Apolónia, 5300-253 Bragança, Portugal; srodrigues@ipb.pt (S.S.R.); teixeira@ipb.pt (A.T.); 4Laboratório Associado para a Sustentabilidade e Tecnologia em Regiões de Montanha (SusTEC), 5300-253 Bragança, Portugal

**Keywords:** salt, meat products, protected designations

## Abstract

The reduction of sodium chloride (NaCl) content, commonly known as salt, in processed meat products is one of the objectives of health organizations and government authorities to achieve healthier products. This reformulation of traditional meat products with protected designations poses more constraints, as they have a more consolidated quality image and less margin for change, since consumers appreciate the products for their unique sensory characteristics. The aim of this work is to present some of the strategies that have been explored to obtain meat products with low sodium content. Information related to the characteristics of traditional meat products with quality marks and geographical indications in different studies is discussed in opposition to the information recorded in their product specifications. It was found that the product specifications of meat products with Portuguese Protected Designation of Origin (PDO) and Protected Geographical Indication (PGI) show a wide variation in the NaCl content, much higher than the recommended values. Thus, one of the requirements to be implemented will be the parameterization of NaCl levels and their monitorization by control and certification organizations as a way to ensure product quality. It is also urgent to examine whether healthy innovation strategies may affect the quality of traditional PDO or PGI meat products and whether they can be included in the respective product specifications.

## 1. Introduction

Meat products are produced and consumed worldwide, but it is in Europe, especially in Mediterranean countries like Portugal, Spain, France, Italy, and Greece, that many of them are considered to have their origin. Several meat products have traditional production methods and are highly appreciated by consumers due to their sensory characteristics, constituting a cultural heritage that is important to preserve. The diversity of meat products is enormous, varying in geographical origins that are often associated with their respective trade names, promoting the development of the regions. Sensory quality is influenced not only by the breed of the animal but also by the climate, formulation, and manufacturing technology, and therefore, the authenticity of these products needs to be recognized with a specific product label. The European Union (EU) quality policy aims to protect specific product names to promote their unique characteristics, linked to their geographical origin as well as traditional know-how. Thus, the Protected Designation of Origin (PDO), Protected Geographical Indication (PGI), and Traditional Specialty Guaranteed (TSG) labels were created associated with meat products and other foods, which are conferred by authorized certifying organizations. Meat products that have received geographical indication recognition are listed as high-quality certified products. These designations include information about each product’s geographic and production specifications, and thus, they should always follow these guidelines.

Currently, in most EU countries, sodium chloride (NaCl) intake (9–12 g/day) is almost three times the amount recommended by the World Health Organization (WHO) and the Food and Agriculture Organization of the United Nations (FAO), which recommend a minimum average daily consumption of 3 g and a maximum of 5 g (equivalent to 2 g of sodium) per person [1]. Salt is composed of 40% sodium and 60% chloride and provides almost 90% of the sodium in the human diet. Although NaCl has an important role in maintaining metabolism through Na^+^ and Cl^−^ actions involved in the regulation and transmission activities of blood, nerves, muscles, and water content in the human body, excessive salt intake represents a risk factor for the onset and/or progression of several chronic diseases, such as hypertension, cardiovascular diseases, and various types of cancer [2,3]. Meat and meat products, for some consumers, have been considered potentially unhealthy foods due to their nutritional characteristics associated with high salt and fat content [4,5,6]. In fact, the high salt and fat content, especially the fat-saturated character of these products, may be a negative characteristic from a nutritional and health point of view. Due to the above, WHO and several government authorities have promoted initiatives to reduce sodium intake by implementing effective measures and policies, including consumer education, awareness campaigns, nutritional labeling, salt content reduction, and food reformulation to promote a healthier diet. These incentive strategies aim to decrease salt content in foods by 25–30% by 2025 [1].

This work intends to highlight the role of salt in meat products and the potential strategies to be implemented without compromising their safety and sensory characteristics. The information related to the characteristics of traditional meat products with quality marks and geographical indications in different studies is discussed, in contrast to the information recorded in their product specifications and which may contradict the health recommendations related to salt intake. Some considerations to be eventually adopted for PDO and PGI meat products are also presented.

## 2. Role of Salt (NaCl) in Meat Products

Salt is considered a multifunctional ingredient for the meat processing industry due to its technological properties. The main functional properties of salt include its water-holding capacity, formation of gels, and emulsion stability through the extraction and solubilization of myofibrillar proteins. Salt improves the adhesion and cohesion of proteins and fat with better mass, which is necessary to acquire the optimal texture of the meat products [7,8,9]. NaCl is also responsible for controlling enzymatic reactions and improving sensory attributes such as color, flavor, and juiciness, as well as the texture of products, stability, and extended shelf life [3,10]. In addition, salt plays an important role as bacteriostatic agent in the safety of sausages by inhibiting the microbiota involved in the deterioration of products as well as the growth of pathogenic bacteria (e.g., *Listeria monocytogenes*, *Clostridium botulinum* and *Staphylococcus aureus*) that can compromise consumers’ health [11,12]. The antimicrobial effect of salt is mainly linked to an increase of the osmotic pressure inside and outside microbial cells and a disruption of the exchange of substances to maintain the normal physiological activities of microorganisms, with a consequent decrease in water activity and restraint in the formation of biogenic amines ensuring the safety of meat products [4,13]. Even though some bacteria are inhibited at low concentrations (2% NaCl), other halotolerant microorganisms are able to develop at high-saline concentrations [14]. In fact, the functional properties of salt in meat products can be compromised with their reduction and/or replacement with other alternative ingredients, such as salt substitutes and flavor enhancers. Therefore, NaCl reduction or replacement must be evaluated, and adjusted if necessary, in order to achieve the desired quality and safety of meat products.

## 3. Strategies for Reducing Salt in Meat Products

The current trend is to offer consumers healthier processed meat products with low salt content but without losing their quality, knowing that reducing or replacing NaCl can have a negative impact on the sensory characteristics of the products [15]. Reducing salt content while preserving safety and sensory properties poses significant challenges. Therefore, despite the technological benefits of NaCl, several integrated strategies have been used to reduce the NaCl content in meat products, namely, reduction by reformulating the products, their total or partial replacement with other non-sodium metal salts, the addition of ingredients, such as aroma enhancers or aromatic herbs, and joint applications with emerging technologies, such as high-pressure processing, pulsed electric field, and ultrasound, to obtain high-quality products with low-sodium content. In fact, the changes observed by the substitution of NaCl in meat products’ sensory characteristics, especially in terms of texture, can be attenuated through the use of maltodextrin, lysine, alanine, citric acid, lactate [16], or glycine [17], maintaining consumer acceptability. The impact of these reformulations on meat products will depend not only on the substitute used but also on the type of meat used and the type of product [18]. Other authors also report changes in the physical form of salt [9,19,20] and improvement of salt diffusion using high-isostatic pressures or ultrasonic waves, radiation, and active packaging technologies, among others [21,22,23]. The use of natural ingredients with potential inhibitory effects on pathogenic microorganisms, such as extracts of essential oils from aromatic plants with a high polyphenol content, has also been suggested as an alternative method to replace or reduce the sodium content in meat products [24]. More recently, the use of edible algae has been referred to as an alternative method to reduce or replace sodium content. Indeed, the market for seaweed-based products is expected to increase in Europe, given current consumer preferences for “new foods”.

### 3.1. Reduction and Replacement of Salts

The processes for direct addition of NaCl are dry salting and brining by immersion or injection. The selection of the type of salt depends on the type of product. In the case of whole pieces, the common method is dry salting, which consists of rubbing or covering the piece with the selected salt [25]. This traditional method could be replaced with brine immersion (10–30%), as this method allows for more effective diffusion [26]. Brine can also be introduced into the piece by injection, as in the case of bacon or ham. In processed meat products, the amount of salt added is mixed with lean meat and the remaining ingredients [27]. One of the most common strategies is the gradual and sustained reduction in the amount of NaCl added to meat products [28] and/or the use of salt substitutes in order to maintain the physicochemical sensory and microbial properties of the product [19]. Examples of salts used as partial substitutes of NaCl are non-chlorinated salts, such as phosphates and lactates [29,30], and chlorinated salts, such as potassium chloride (KCl), calcium chloride (CaCl_2_), and magnesium chloride (MgCl_2_) [31]. Both sodium and potassium phosphates and lactates have been used as alternatives to salt. However, the extent of sodium reduction depends on the phosphates and lactates used in meat products. In general, lower concentrations of sodium lactate are required to prevent bacterial proliferation compared to salt [32], most likely because lactates reduce the water activity in food more than salt. At the industrial level, potassium chloride (KCl) has been commonly used and recognized as GRAS (Generally Recognized as Safe) due to its molecular similarity to NaCl [19,33]. According to Bidlas and Lambert [34], KCl has an antimicrobial effect equivalent to that of NaCl on certain microorganisms, such as Aeromonas hydrophila, Enterobacter sakazakii, Shigella flexneri, Yersinia enterocolitica, and Staphylococcus aureus. The replacement of NaCl with KCl makes it possible to safely obtain meat products with low sodium content. However, KCl in large amounts can impart a bitter and metallic taste to meat products. Conversely, Neves and colleagues [35] observed that the replacement of NaCl with KCl did not affect the perception of saltiness and the acceptability of pork or ham, even when pale, soft, and exudative meats were used. Several studies refer to the use of KCl as a salt substitute [36,37], and they found that the replacement of NaCl with KCl up to 30% improves the rheological and sensory acceptability of the product without any increased risk to the consumer’s health [38]. Some authors have used KCl and CaCl_2_ as salt substitutes in meat products [19,25,39]. The replacement of NaCl with a mixture of potassium/magnesium/calcium in meat products also favors the enrichment of the product in calcium and magnesium salts [14,26]. Table 1 shows some strategies for reducing and replacing NaCl in meat products and their effect on physicochemical parameters, sensory characteristics, and microbiota. These studies demonstrate how the amount of each salt used is essential to the control of biochemical reactions (e.g., lipolysis and proteolysis), antimicrobial activity, and to avoid the appearance of technological and sensory changes that can cause product depreciation [18,40]. In fact, the replacement of NaCl with other salts may induce lipid oxidation with production of volatile compounds, such as hydrocarbons and aldehydes [41]. According to Flores [42], the reduction of NaCl increases the concentration of aldehydes (hexanal, heptanal, 2-nonenal, pentanal) producing unpleasant herbal odors.

Besides an increase of volatile compounds in fermented/dry-cured sausages [43,45], several studies have also reported high levels of biogenic amines with NaCl reduction, without compromising product safety [7,46]. These harmful substances are mainly formed by the decarboxylation of amino acid precursors through microbial decarboxylases. Contrarily, the NaCl replacement with non-sodium metal salts, such as KCl, can reduce the amounts of putrescine, cadaverine, and histamine in bacon [54]. The use of other chloride salts (KCl, CaCl_2_, and MgCl_2_) in cured meat products can also modify the activity of some endogenous enzymes, intensifying proteolysis reactions with an increase in the amount of free amino acids, mainly leucine, valine, alanine, and phenylalanine [25,41]. These free amino acids, in particular, can have an impact on sensory properties, as their concentration is associated with bitter tastes. Thus, the addition of these salts to meat products should be in limited amounts due to their bitter, metallic, and astringent tastes, which can also influence the textural properties and result in rejection of the product by consumer [55,58]. Studies carried out on dry-cured meat products have shown that the substitution of NaCl with a mixture of salts (KCl, CaCl_2_, and MgCl_2_) in percentages greater than 25% leads to changes in aroma and taste, color, texture, and in general, a lower acceptance of the products by consumers [59]. It is often necessary to use flavor enhancers to “mask”, confer, and/or enhance the taste and/or smell of food, thus, mitigating some of the undesirable sensory effects [60]. Some flavor and taste enhancers, like glutamates, yeast extracts, 5’-nucleotides, amino acids, organic acids, and vegetable protein hydrolysates can be added to balance the salty taste of food products with lower salt content [10,18,38,45,56,61]. As shown in Table 1, salt reduction and/or replacement also has an effective role in the microbial stability of meat products [43,46,47]. Elias et al. [47] evaluated the impact of a 25% salt reduction on traditional dry-cured sausage on its microbiological stability. The authors observed that *E. coli*, *Salmonella* spp. and *Listeria monocytogenes* were not present and the technological microbiota, LAB, and enterococci, which are involved in meat fermentation, achieved high counts in the final product. Also, Michelakou and colleagues [50] found that a 50% NaCl reduction on traditional cooked and smoked Greek pork meat products remained microbiologically stable under refrigerated vacuum storage for approximately two weeks, being finally spoiled by Brochothrix thermosphacta, which results in the deterioration of taste, odor, and overall appearance of the product.

### 3.2. Use of Alternative Ingredients

An alternative to the use of salt in meat products is the use of natural ingredients, i.e., products that are free of synthetic additives. In this context, the reduction and/or replacement of NaCl content in meat products can be obtained by a combination with other ingredients, such as aromatic herbs, essential oils, yeast extracts [61,62], and seaweeds [57,63,64], among others. The use of these umami-tasting compounds has been increasingly common in the food industry since they have nutritional properties and enhance the flavor of food without requiring the addition of salt in great amounts that may be harmful to health. Many studies have been conducted to explore the effects of salt reduction in meat products by incorporating aromatic herbs (e.g., rosemary, basil, and oregano) and halophytes extracts (belonging to *Salicornia* and *Sarcocornia* genera) simultaneously imparting flavor and taste [24]. Kohri et al. [65] found that adding 0.35% herbal extracts (hot water extracts of basil, rosemary, parsley, fennel, and oregano) increase the saltiness by 1.13 to 1.22 times and are expected to reduce salt usage by about 10% to 20%. These alternatives, in addition to functioning as flavor enhancers, can also influence color and texture, and act as antimicrobial agents and antioxidants. These NaCl alternative ingredients are rich in polyphenols and flavonoids that play a significant role in neutralizing free radicals and inhibiting lipid oxidation, thereby helping to preserve the quality of low-salt meat products, minimizing oxidative spoilage, and extending shelf life. Essential oils from spices, used individually or in combination, are highly inhibitory against spoilage and pathogenic microorganisms [66]. For labeling purposes, and according to Regulation (EU) No 1169/2011 [67], there are no regulated acceptable maximum limits for the use of this type of ingredient.

Several studies also mentioned the utilization of yeast extracts [61] and edible algae as interesting possibilities to develop new low-salt meat products [57,64,68]. Yeast extracts are also widely used in the food industry and play an important role in contributing to the enhancement and improvement of flavor. Yeast extracts can even have a synergistic effect with other ingredients on taste [61,69]. According to Delgado-Pando and colleagues [62], the manufacture of low-salt ham was optimized by using a mixture of glycine and yeast extract as flavorings. This combination enhances the flavor of low-salt dry-cured ham due to the synergistic effect between the umami compounds in yeast extract and salt as well as the taste supplied by glycine. The authors found that it is possible to produce ham with up to a 20% salt reduction, and the concomitant use of a mixture of 0.4% yeast extract and/or glycine without impairing the acceptability of the product by the consumer and the need to add high amounts of salt substitutes to maintain the quality of the product. In line, Vidal et al. [70] reported that the incorporation of lysine and yeast extracts in low-sodium salted meat improved sensory acceptance and reduced rancidity and saltiness, using a curing mixture of NaCl + KCl + CaCl_2_.

The use of seaweeds as a natural ingredient and/or food supplement is also an alternative that has been explored, given its promising benefits for human health. Seaweeds are a great source of nutrients and bioactive compounds, such as protein, polysaccharides, omega-3 fatty acids, phenolic compounds, carotenoids, vitamins, and minerals [64]. Due to their mineral content and composition, seaweeds are an excellent alternative method of formulating low-sodium meat products. Despite showing benefits due to its nutritional and physicochemical properties, the application of edible algae in meat and meat products to replace NaCl is still at an early stage [57,64]. Vilar et al. [57] evaluated the inclusion of 1% of two red (*Porphyra umbilicalis* and *P. palmate*) and two brown (*Himanthalia elongata* and *Undaria pinnatifida*) edible seaweeds in reformulated frankfurters with a 50% reduction of salt. Significant differences in color, aroma, flavor, and texture attributes were evident in reformulated frankfurters. Moreover, the overall acceptability of seaweed-frankfurters was greatly influenced by the type of added seaweed. In the study performed by Barbieri et al. [71], a reduction of the salt content (1.8% to 1.2% or 1.0%) in ham was achieved by modifying the thermal process and adding water soluble extract of *Palmaria palmata* as a salt replacer and flavoring, thus reducing sodium intake by 25% and 35%, respectively. Although the incorporation of seaweeds in meat products could be effective, further work is required to optimize dose rates to enhance safety, physicochemical, and sensory properties as well as consumer acceptance. 

It remains important that reduction and/or replacement of sodium with these substitutes in processed meat products could result in consumers not accepting some of the product reformulation outcomes. On the other hand, reducing and/or replacing meat products with alternative substitutes resulted in reduced intakes of NaCl and has the potential to facilitate the shift towards diets lower in salt associated with health benefits. Consumption of low-sodium meat products can contribute to reduced blood pressure and reduced incidence of various cancers, intestinal defects, and cardiac and lung diseases. Therefore, these findings are significant for analyzing the potential health effects of reducing and/or replacing sodium salt with alternative ingredients.

## 4. PDO and PGI Meat Products: The Portuguese Case Study

The quality of a meat product is a complex and multidimensional concept. It involves considering the specifications of each product and the characteristics that meet the needs, preferences, and tastes of consumers, while also addressing health concerns. Therefore, we can define the quality of a meat product as a combination of quantitative and qualitative characteristics whose relative importance maximizes consumer acceptance and economic value in relation to market demand. The EU’s quality policy aims to assure the names of certain products, emphasizing their unique qualities tied to geographical origin and traditional expertise. According to the European Commission Council Regulation (EC) No 510/2006 [72], Protected Designation of Origin (PDO) is understood as a name that identifies a product originating from a specific place or region (or, in exceptional cases, from a country), whose quality or characteristics are essential or exclusive to a specific geographic area (including its natural and human factors), and whose production phases take place in it. Protected Geographical Indication (PGI) is understood as a name that identifies a product originating from a specific place or region (or a country), which has a quality, reputation, or other characteristics that can be essentially attributed to its geographical origin and in relation to which at least one of the production phases takes place in the delimited geographical area. Products with PDO and PGI names on the label have an image associated with better quality, positive sensory attributes, and a strong association with a particular origin or location for consumers. For the consumer, a quality product is one that offers the most pleasant sensory characteristics while being as healthy as possible. While sensory characteristics are subjective and can vary from person to person, health characteristics depend primarily on the quality of the raw materials, salt concentration, the use of additives, and the processing methods, such as fermentation, drying, and smoking. Ensuring the preservation and healthiness of products is crucial, with a particular focus on the health issues caused by sodium consumption through the NaCl present in food products.

Currently, the European Union strives to achieve sustainable development goals in alignment with United Nations policies [73]. To support this vision, the EU is working to incorporate sustainable practices into the quality schemes of PDO and PGI for food products. By recognizing sustainable practices, producers can enhance the value of their actions concerning environmental (good resources use, minimum waste, circular economy, maintenance of biodiversity, good management of cultural heritage, life cycle assessment), economic (business support, job creation, and maintenance with development of rural areas), and social sustainable practices including animal welfare. PDO and PGI meat products can be highly valued and have a high impact on the social life of communities. They also serve as excellent examples of local production utilizing sustainable practices. It is essential to incorporate this vision into their processes and business models to communicate these values to consumers effectively. Additionally, efforts should be made to reduce the health impacts of these products, such as lowering their salt content. The current trend in food consumption advocates for reducing salt intake due to the well-documented link between excessive salt consumption and cardiovascular disease, high blood pressure, diabetes, and kidney disease. Public health entities and organizations are increasingly focused on controlling salt consumption. According to Cobiac et al. [74], reducing sodium intake to less than 5 to 6 g per day would benefit consumers’ health and lower healthcare costs. In this revision, we focus on Portuguese PDO and PGI meat product specifications registered in the EU regarding salt content (Table 2). These products generally include semi-dry smoked fermented sausages that require heat treatment before consumption, dry smoked cured sausages, and cured and dried meat pieces that can be consumed without culinary preparation. Table 2 reveals that the specifications for PDO and PGI products globally exhibit significant variations in their salt content, with some products containing salt percentages above 4% and up to 10–11%. Comparatively to other Mediterranean countries where the production of traditional meat products has been renewed, similar products from Spain, Italy, and France are presented as well, along with their NaCl maximum % values: in Spain, the Jamón de Trevélez has a salt value lower than 5% (PGI-ES-0309) while in Italy, the Prosciutto di Parma (PDO-IT-0067) ranges from 4.2 to 6.0%, and in France, the specification of Jambon d’Auvergne presented salt values < to 6.5%. Regarding dry-cured sausages, France, Italy (Salame Piemonte, PGI-IT-1237, 3.8%), and Spain (Chorizo de Cantimpalos, PGI-ES-0632, <6%) described their products, but the reference to the salt content is voluntary with maximum levels very similar to the Portuguese ones. The salt content in PDO and PGI meat products from different Mediterranean countries is above the most recent recommendations from health organizations, which advocate for reducing salt content in foods. The salt levels in the specifications have indications for maximum values that cannot be surpassed; however, lower values can be used if the safety and quality are not compromised. The aim of the certification bodies is to ensure compliance of the meat products with the requirements stated, achieving the level of quality defined. So, it will be possible to use lower salt content in PDO and PGI meat products without contradicting their specifications. 

Table 3 presents the salt contents in some Portuguese PDO and PGI meat products characterized by several studies. 

Patarata et al. [75] conducted a study on the sensory and physicochemical characteristics of alheiras sourced from 40 different producers, including both small-scale operations (27 samples) and industrial-scale producers (13 samples). The salt content varied from 1.0% to 3.2%, likely due to the different formulations used by small producers and larger industries. However, no negative comments were noted regarding the saltiness of the product.

The effect of partially replacing NaCl with KCl and Sub4Salt^®^ on the quality of Bisaro pork sausages (PGI *Chouriça de Vinhais*) was investigated [80]. Three different formulations (NaCl + KCl, NaCl + Sub4Salt^®^, and KCl + Sub4Salt^®^) were compared to the control (2% NaCl). The authors concluded that replacing NaCl did not affect the pH, water activity (aw), or the chemical, microbiological, and sensory qualities of the sausages, suggesting that this is a viable strategy for the meat industry to produce “reduced sodium content” sausages without compromising their traditional quality.

The influence of salt content on the presence and production of biogenic amines (BA) in *Painho de Portalegre* was also studied [82]. The authors increased the salt content to 6% in some batches and reduced it to 3% in others, while the typical formulations used have a final content of 4%. They concluded that salt content influences BA production due to its effect on the microbial population. The effect of salt reduction of meat products could be contradictory and depends on several factors apart from salt reduction, namely, microbial quality of raw materials, processing hygiene, and final product stability [87]. 

Generally, the salt contents in the Portuguese meat products studied (Table 3) are within or even below the values indicated by their respective specifications, as seen in the cases of fermented dry sausages *Alheira* and *Chouriço*. Exceptions include dry-cured pieces *Paio do Alentejo* and *Painho de Portalegre*, likely due to their larger size and longer curing process. These scientific findings suggest that it is feasible to achieve lower salt content in PDO and PGI meat products. 

Reducing the salt content in processed meat products is one of the main strategies for obtaining healthier products [88], although this should be conducted progressively without risks to product safety [87]. It is mandatory to study the impact of salt reduction on safety, shelf life, and sensory characteristics of PDO and PGI meat products using strategies of salt reduction, salt replacement with other salts (KCl, CaCl_2_ or MgCl_2_), and even using spices and aromatic herbs in use and according to tradition know-how. The reduction of salt or its replacement does not counter the actual specifications in place and registered in the EU. Even so, it will be possible to propose and admit changes to actual specifications only if there is evidence reported.

## 5. Conclusions and Future Perspectives

Salt has been used as a preservative agent in human diets since ancient times. Approximately 20% of the total sodium intake derives from meat products that contain high NaCl content, though excessive intake of sodium salt poses health concerns. It is crucial for the meat industry to reduce the amounts of salt added during processing of meat products. This review summarizes different strategies to reduce and replace the salt content in meat products. The impact of non-sodium salt alternatives and natural ingredients on the physicochemical, sensory, and microbial properties of meat products were discussed. The achievement of low-sodium meat products presents great challenges, such as sensory attributes (color, taste, texture, and flavor), microbial safety, applicability, and consumer acceptance. Reformulation of meat products with alternative ingredients, alone or in combination, could be a promising approach to obtaining reduced-salt products. 

Since the PDO and PGI meat products characterized show a wide variation in NaCl content, often exceeding the recommended values, it is urgent to reduce this content without changing their sensory characteristics and safety. The application of different strategies to PDO and PGI meat products such as simple reduction or salt replacement is possible without contradicting the actual specifications. Therefore, reducing the salt content is voluntary but necessary, and consequently, salt reduction or replacement strategies must ensure that sensory quality and safety are not compromised. Moreover, educational approaches of producers should be taken. Scientific evidence will support proposing changes to the next EU quality scheme specifications and integrating sustainable goals into their processes.

## Figures and Tables

**Table 1 foods-13-02681-t001:** Effects of salt reduction and/or replacement in meat products.

Strategies/Product Type	Name of Product	Process	Effects			Ref.
			Physicochemical	Sensory/Technological	Microbiological	
**NaCl reduction**						
Fermented/dry-cured sausages	*Chorizo Cantimpalos*	16% NaCl reduction (2.7% to 2.26%)	- decrease sulfur compounds concentration - increase aldehyde content	- decrease of aroma, flavor, and juiciness - decrease cohesion		[43]
		17–20% NaCl reduction with *Debaryomyces hansenii* inoculation	- increase water activity	- improve of aroma and taste - increase hardness and chewiness	- decrease growth of staphylococci	[44]
		50% NaCl reduction (2.5% to 1.3%)	- increase volatile compounds concentration - increase total free amino acids	- decrease firmness		[45]
	*Catalão* and *Salsichão*	50% NaCl reduction (6% to 3%)	- high biogenic amines (histamine, tyramine, and cadaverine) concentration with 3% salt	- less intense aroma in products with 3% salt	- salt reduction does not compromise product safety	[46]
		25% NaCl reduction (4% to 3%)	- no effect on pH and water activity	- salt reduction affects aroma (less aromatic) - more adhesive and less cohesive and resilient	- no effect on microbiota	[47]
	*Sliced chouriço*	50% NaCl reduction (3% to 1.5%)			- increase survival rate of *L. monocytogenes*	[48]
	*Chouriço preto and Paio preto*	50% NaCl reduction (6% to 3%)	- high biogenic amines content	- no changes in texture (sensory panel)		[7]
Cured, cooked and/or smoked pieces	*Bacon*	13% NaCl reduction (2.9% to 2.5%, 2.0% and 1.5%)	- no changes on pH and water activity	- increase of hardness	- high total viable counts, mainly LAB	[49]
	*Ham*	20% NaCl reduction (1.3% to 1.27%)	- no changes on pH and water activity	- increase of hardness and chewiness	- high total viable counts, mainly LAB, during storage	
	*Syglino of Monemvasia*	50% NaCl reduction (4.9% to 2.5%)		- deterioration of taste, odor and overall acceptance after 2 weeks	- microbiologically stable under refrigerated vacuum storage - grown of *Brochothrix thermosphacta* after 2 weeks	[50]
**NaCl replacement with other non-sodium metal salts**						
Dry fermented/cured sausages	*Chorizos*	50% NaCl replaced by K, Mg and Ca ascorbates		- less consistency and lower acceptability (less salty taste)		[51,52]
	*Salame*	NaCl replaced by mixture of KCl and CaCl_2_	- 55% decrease in Na content	- no changes on sensory properties		[53]
	*Slow fermented sausage*	2.7% NaCl replaced by 2.26% NaCl +0.43% KCl	- increase in aldehyde compounds	- the same acceptability by consumers as for control, except for aroma that was not improved by KCl addition	- no changes on LAB and Staphylococci	[43]
		25% and 50% NaCl replaced by KCl and CaCl_2_	- increase of lipid oxidation during processing and storage			[45]
Cured, cooked and/or smoked pieces	*Lacón*	50%NaCl + 50% KCl; 45%NaCl + 25% KCl + 20% CaCl_2_ +10% MgCl_2_; 30%NaCl + 50%KCl + 15% CaCl_2_ + 5% MgCl_2_	- high free amino acids content			[25]
	*Cecina*	50% NaCl + 50% KCl; 45%NaCl + 25% KCl + 20% CaCl_2_ +10% MgCl_2_	- 50% KCl increases protein content and decreases moisture; - substitution with CaCl_2_ and MgCl_2_ increases lipid oxidation and luminosity (L*), but decreases red (a*) and mineral (Na) content	- no changes in texture		[26]
	*Bacon*	80%NaCl + 20%KCl 60% NaCl + 40% KCl 40% NaCl + 60% KCl	- no changes in humidity, pH, and total volatile basic nitrogen (ABVT); - 40% NaCL decreases putrescine, cadaverine, and histamine levels; - 60% KCl increases N nitrosodimethylamine (NDMA) content	- 40% NaCl and less amount does not alter color and aroma		[54]
**NaCl replacement with other non-sodium metal salts and alternative ingredients**						
Fermented sausages		replacement with 0–60% KCl, 0–100% potassium lactate and 0–100% glycine	- slow decrease in pH	- more than 40% KCl imparts bitter taste - more than 40% lactate potassium alters texture, color and taste		[55]
		replacement of 25% or 50% NaCl by KCl and supplementation with 1–2% yeast extract (*Saccharomyces cerevisiae*)		- 50% replacement with KCl decreases sensory quality; 2% extract supplementation increases volatile compounds		[56]
Cured, cooked sausages	*Salsichas Frankfurter*	1% NaCl and 1% replacement with edible algae (*Porphyra umbilicalis*, *P. palmate*, *Himanthalia elongata*, and *Undaria pinnatifida*)	- decrease in color parameters: luminosity (L*), red (a*) and yellow (b*)	- the intense flavor of the seaweed negatively affects the overall acceptability		[57]
		0–65% NaCl 35–100% KCl 0–20% glycine	- no changes on color	- more than 80% KCl promotes bitter taste		[18]

**Table 2 foods-13-02681-t002:** Percentage of salt in Portuguese meat products with PDO or PGI protection, with indication of producer group and Control and Certification Organism (CCO).

Product Type	Name Product	Protection	Producer Association ^1^	Control and Certification Organism ^2^ (CCO)	% Salt
Semi-dry smoked, fermented sausage	*Alheira de Barroso-Montalegre*	PGI	---	No CCO	Maximum 4%
	*Alheira de Mirandela*	PGI	ACIM	*Tradição e Qualidade*	No mention
	*Alheira de Vinhais*	PGI	ANCSUB	*Tradição e Qualidade*	Maximum 4%
	*Chouriça Doce de Vinhais*	PGI	ANCSUB	*Tradição e Qualidade*	Maximum 3%
	*Chouriço Azedo de Vinhais/Azedo de Vinhais/Chouriço de Pão de Vinhais*	PGI	ANCSUB	*Tradição e Qualidade*	Maximum 3%
	*Chouriço de Abóbora de Barroso-Montalegre*	PGI	---	No CCO	Maximum 4%
Cured smoked semi-dry sausage	*Morcela de Assar de Portalegre*	PGI	NATUR-AL-CARNES	No CCO	Maximum 4%
	*Morcela de Estremoz e Borba*	PGI	APETAL	No CCO	2–4%
Dry smoked, cured sausage	*Butelo de Vinhais/Bucho de Vinhais/Chouriço de Ossos de Vinhais*	PGI	ANCSUB	*Tradição e Qualidade*	No mention
	*Chouriça de Carne de Barroso-Montalegre*	PGI	---	No CCO	Maximum 4%
	*Chouriça de Carne de Melgaço*	PGI	MELGAÇO RURAL	Kiwa Sativa	Maximum 6%
	*Chouriça de Carne de Vinhais/Linguiça de Vinhais*	PGI	ANCSUB	*Tradição e Qualidade*	Maximum 6%
	*Chouriça de Sangue de Melgaço*	PGI	MELGAÇO RURAL	Kiwa Sativa	Maximum 3%
	*Chouriço de Carne de Estremoz e Borba*	PGI	APETAL	No CCO	Maximum 6%
	*Chouriço de Portalegre*	PGI	NATUR-AL-CARNES	No CCO	4–6%
	*Chouriço Grosso de Estremoz e Borba*	PGI	APETAL	No CCO	Maximum 6%
	*Chouriço Mouro de Portalegre*	PGI	NATUR-AL-CARNES	No CCO	2–4%
	*Linguiça de Portalegre*	PGI	NATUR-AL-CARNES	No CCO	No mention
	*Linguíça do Baixo Alentejo/Chouriço de carne do Baixo Alentejo*	PGI	ANCPA	No CCO	4–6%
Cured/matured, smoked semi-dry sausages	*Farinheira de Estremoz e Borba*	PGI	APETAL	No CCO	5–6%
	*Farinheira de Portalegre*	PGI	NATUR-AL-CARNES	No CCO	5–6%
	*Sangueira de Barroso-Montalegre*	PGI	---	No CCO	Maximum 4%
Blanched sausage	*Morcela de cozer de Portalegre*	PGI	NATUR-AL-CARNES	No CCO	Maximum 6%
Blanched marinated sausage	*Cacholeira branca de Portalegre*	PGI	NATUR-AL-CARNES	No CCO	Maximum 6%
Dry smoked, cured meat pieces	*Presunto de Barrancos/Paleta de Barrancos*	PDO	ACPA	AGRICERT	6–7%
	*Presunto de Barroso*	PGI	---	No CCO	No mention
	*Presunto de Melgaço*	PGI	MELGAÇO RURAL	Kiwa Sativa	8–10%
	*Presunto de Vinhais/Presunto Bísaro de Vinhais*	PGI	ANCSUB	*Tradição e Qualidade*	9–11%
	*Presunto do Alentejo/Paleta do Alentejo*	PDO	ACPA	AGRICERT	No mention
	*Presunto de Campo Maior e Elvas/Paleta de Campo Maior e Elvas*	PGI	ACPA	No CCO	No mention
	*Presunto de Santana da Serra/Paleta de Santana da Serra*	PGI	ACPA	AGRICERT	No mention
Dry smoked cured bagged meat pieces	*Lombo branco de Portalegre*	PGI	NATUR-AL-CARNES	No CCO	5–7%
	*Lombo enguitado de Portalegre*	PGI	NATUR-AL-CARNES	No CCO	5–7%
	*Salpicão de Barroso-Montalegre*	PGI	---	No CCO	Maximum 4%
	*Salpicão de Melgaço*	PGI	MELGAÇO RURAL	Kiwa Sativa	2–4%
	*Salpicão de Vinhais*	PGI	ANCSUB	*Tradição e Qualidade*	Maximum 5%
	*Paia de Estremoz e Borba*	PGI	APETAL	No CCO	Maximum 6%
	*Paia de lombo de Estremoz e Borba*	PGI	APETAL	No CCO	Maximum 7%
	*Paia de toucinho de Estremoz e Borba*	PGI	APETAL	No CCO	Maximum 7%
	*Painho de Portalegre*	PGI	NATUR-AL-CARNES	No CCO	5–6%
	*Paio de Beja*	PGI	ANCPA	No CCO	4–6%

**^1^** ACIM—*Associação Comercial e Industrial de Mirandela*; ACPA—*Associação de Criadores de Porco Alentejano*; ANCPA—*Associação Nacional de Criadores de Porco Alentejano*; ANCSUB—*Associação Nacional de Criadores de Suínos da Raça Bísara*; APETAL—*Agrupamento de Produtores de Enchidos Tradicionais da Alentejo, Lda.*; CAB—*Cooperativa Agrícola de Beja*; MELGAÇO RURAL—*Associação de Produtores Locais*; NATUR-AL-CARNES—*Agrupamento de Produtores Pecuários do Norte Alentejano, S.A.* ^2^ AGRICERT—*Certificação de Produtos Alimentares, Lda.*; CERTIS—*Controlo e Certificação, Lda.*

**Table 3 foods-13-02681-t003:** Percentage of salt in some Portuguese meat products with PDO and PGI protection.

Product	% Salt	References
*Alheira*	1.0–3.2	[75]
	1.3–1.5	[76]
	1.6–1.9	[77]
	1.0–1.3	[78]
*Chouriça Vinhais*	1.3−2.9	[79]
	1.5−2.0	[80]
*Chouriça Carne Melgaço*	2.4–3.7	[81]
*Salpicão*	1.5–4.3	[79]
*Painho de Portalegre*	3.0–6.0	[82]
	2.5	[83]
*Chouriço grosso de Estremoz e Borba*	1.3	[84]
*Paio do Alentejo*	2.9–6.0	[85]
*Presunto Bísaro*	6.1–6.9	[86]

## Data Availability

No new data were created or analyzed in this study. Data sharing is not applicable to this article.

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
