# Peer review of "Strategies to Reduce Salt Content: PDO and PGI Meat Products Case"

_foods, 2024, doi:10.3390/foods13172681_

Round 1

Reviewer 1 Report

Comments and Suggestions for Authors

The topic of the paper is very interesting and timely. However, I have several major concerns that need to be addressed or amended.

Is it possible to replace the salt content with some alternative substitutes if the product is protected? I am referring to its original version in the specification. If yes, what procedure should be followed? This part should definitely be elaborated in the paper.

What should motivate producers to change their recipes and start the recertification process of their product? Personally, I do not see the point after the lengthy process they have gone through, unless the EU steps in, funds everything—from initiating the process, making recommendations, providing education, to financing, etc.

The subsection on alternative ingredients is written too generally. It should be detailed and clearly discussed—describing the procedures for substituting alternative ingredients, consumer acceptance, comparisons of individual products, etc.

You described the Portuguese case of PDO and PGI products; in that subsection, there is no discussion related to the issues faced by individual producers, why the salt content was reduced, how it was done, how it aligns with EU regulations, why some producers did this, and whether it is punishable considering it does not comply with the specification?

The conclusion is written too generally; everything mentioned is known without your study. What detailed strategies need to be initiated to reduce the salt content in PDO and PGI products?

Author Response

Response to the reviewers' comments:

The authors thank the availability of the reviewers to review and comment on their manuscript. We tried to address all the concerns as described below. We have made the corrections and included the suggestions in the revised form of our manuscript.

Reviewer #1: The topic of the paper is very interesting and timely. However, I have several major concerns that need to be addressed or amended.

- Is it possible to replace the salt content with some alternative substitutes if the product is protected? I am referring to its original version in the specification. If yes, what procedure should be followed? This part should definitely be elaborated in the paper.

Reply: The authors truly believe so, since the specifications for PDO or PGI products do not specify what type of salt, especially if it is a question of reducing or replacing the sodium ion, for example, with potassium. Examples of reducing and replacing sodium in PDO or PGI products were described in the body text and this opinion was also stated (see pages 12 and 13).  

- What should motivate producers to change their recipes and start the recertification process of their product? Personally, I do not see the point after the lengthy process they have gone through, unless the EU steps in, funds everything—from initiating the process, making recommendations, providing education, to financing, etc.

Reply: When it comes to seeking a healthier product by reducing the salt content, we see no reason why it shouldn't motivate producers. What's more, the specifications most often limit the amount of salt, and do not impose a minimum quantity. Producers associations should be attentive to the trends related with minimizing salt content or even additives, and promote recommendations and educative actions and even future recertification processes.  

- The subsection on alternative ingredients is written too generally. It should be detailed and clearly discussed—describing the procedures for substituting alternative ingredients, consumer acceptance, comparisons of individual products, etc.

Reply: The authors have discussed more detailed the subsection of alternative ingredients, as requested by the reviewer (see pages 7 and 8).

- You described the Portuguese case of PDO and PGI products; in that subsection, there is no discussion related to the issues faced by individual producers, why the salt content was reduced, how it was done, how it aligns with EU regulations, why some producers did this, and whether it is punishable considering it does not comply with the specification?

Reply: The descriptions deal with experimental work that could motivate the industry to adopt strategies to reduce the salt content of meat products for obvious reasons related to combating heart and vascular diseases. According to the reviewer request a better description was added, and for the reduction or salt replacement was stated that there is no punishment.

- The conclusion is written too generally; everything mentioned is known without your study. What detailed strategies need to be initiated to reduce the salt content in PDO and PGI products?

Reply: Detailed strategies for salt reduction and replacement was described supported by studies already done; however according to different PDO or PGI meat products, more studies should be done. Statements related with these ideas was written in the text.

Reviewer 2 Report

Comments and Suggestions for Authors

Here are some of my comments and questions about this manuscript.

- Impact of Salt Reduction on Microbial Safety: While the study discusses the sensory and physicochemical properties, it lacks an in-depth analysis of the microbial safety of reduced-salt products. Including this discussion would provide a more comprehensive evaluation.

- Future Research Directions: The study lacks a dedicated section on future research directions. The authors should suggest specific areas for further investigation based on their findings.

- Limitations and Bias: The paper does not address potential limitations or biases in the study. The authors should discuss any limitations or sources of bias and how they were mitigated.

- Effectiveness of Replacement Ingredients: The paper discusses various salt replacements but does not evaluate their effectiveness comprehensively. A more detailed analysis of how each replacement ingredient impacts the final product would be beneficial.

- Consumer Health Impact: The discussion on the health benefits of reduced salt content is limited. A more detailed analysis of how these changes impact consumer health would strengthen the paper.

- Regulatory Implications: The paper mentions the need for regulatory changes but does not provide detailed recommendations. The authors should discuss potential regulatory implications and how their findings could inform policy changes.

- Use of Visual Aids: The paper could benefit from more visual aids, such as graphs and charts, to present data more effectively. This would help in better understanding the results and comparisons made.

Author Response

Reviewer #2: Here are some of my comments and questions about this manuscript.

- Impact of Salt Reduction on Microbial Safety: While the study discusses the sensory and physicochemical properties, it lacks an in-depth analysis of the microbial safety of reduced-salt products. Including this discussion would provide a more comprehensive evaluation.

Reply: The authors acknowledge the reviewer remark and additional information about the microbial safety was added in the manuscript.

- Future Research Directions: The study lacks a dedicated section on future research directions. The authors should suggest specific areas for further investigation based on their findings.

Reply: The reviewer’s point of view is totally correct. The authors reformulated the last part of the conclusion by including future directions, as requested by the reviewer.

- Limitations and Bias: The paper does not address potential limitations or biases in the study. The authors should discuss any limitations or sources of bias and how they were mitigated.

Reply: The authors acknowledge the reviewer suggestion. Potential limitation of the strategies adopted was added in the text (see page 8).

- Effectiveness of Replacement Ingredients: The paper discusses various salt replacements but does not evaluate their effectiveness comprehensively. A more detailed analysis of how each replacement ingredient impacts the final product would be beneficial.

Reply: The authors acknowledge the reviewer remark, and the discussion was modified in the manuscript according with the reviewer suggestion.

- Consumer Health Impact: The discussion on the health benefits of reduced salt content is limited. A more detailed analysis of how these changes impact consumer health would strengthen the paper.

Reply: The changes of reduction of replacing salt in meat products were analyzed regarding health benefits and included in the review, see page 8.

- Regulatory Implications: The paper mentions the need for regulatory changes but does not provide detailed recommendations. The authors should discuss potential regulatory implications and how their findings could inform policy changes.

Reply: Clarification about the implications of reduction and salt replacement in the specifications of PDO and PGI meat products were done. Future findings regarding the impact of alternative salt replacements in these products will be necessary to inform and shape EU policies for potential updates to the current labeling quality schemes

- Use of Visual Aids: The paper could benefit from more visual aids, such as graphs and charts, to present data more effectively. This would help in better understanding the results and comparisons made.

Reply: Sorry we do not have any additional visual aids.

Reviewer 3 Report

Comments and Suggestions for Authors

In this review, the critical issue of reducing sodium chloride (NaCl) content in processed meat products, particularly those with Protected Designation of Origin (PDO) and Protected Geographical Indication (PGI) status were summarizes.The overall content of the article is relatively substantial. There are some problems must be solved. So I suggest minor revision.

1. Page 2, Lines 49-61. It is recommended to increase the harm of high salt diet, as well as information on the salt and fat content of meat with different labels.

2. Page 7, Lines 206. The strategies for reducing salt in meat products had been  listed. It is better to increase the effectiveness of comparison between different strategies.

3. Page 7, Lines 207. The discussion on PDO and PGI meat products in the Portuguese case study. The ability to maintain product quality while reducing salt content should be added

4. Page 11, Line 264. Is there any reference about the critical review?

5. Page 11, Line 278-285. What is the specific situation and classification of PDO and PGI meat products in various EU countries?

Author Response

Reviewer #3: In this review, the critical issue of reducing sodium chloride (NaCl) content in processed meat products, particularly those with Protected Designation of Origin (PDO) and Protected Geographical Indication (PGI) status were summarizes. The overall content of the article is relatively substantial. There are some problems, which must be solved before it is considered for publication. So I suggest minor revision.

  1. Page 2, Lines 49-61. It is recommended to increase the harm of high salt diet, as well as information on the salt and fat content of meat with different labels.

Reply: The authors acknowledge the reviewer remark. This information was added in page 2, lines 63-65; the salt content of different label PDO and PGI meat products was described on tables 2 and 3.

  1. Page 7, Lines 206. The strategies for reducing salt in meat products had been listed. It is better to increase the effectiveness of comparison between different strategies.

Reply: Corrected, as requested by the reviewer.

  1. Page 7, Lines 207. The discussion on PDO and PGI meat products in the Portuguese case study. The ability to maintain product quality while reducing salt content should be added

Reply: Comments regarding how the stability of the products can be reached while reducing salt were done.

  1. Page 11, Line 264. Is there any reference about the critical review?

Reply: The authors have changed the sentence of the manuscript in page 9, lines 394-396.

  1. Page 11, Line 278-285. What is the specific situation and classification of PDO and PGI meat products in various EU countries?

Reply: Clarification regarding specific situation and classification of PDO and PGI meat products in various EU countries was added.

Round 2

Reviewer 1 Report

Comments and Suggestions for Authors

Dear Authors,

Thank you for the thorough responses you provided. 

In addition, I have a few small suggestions that I hope you will find helpful.

l63 which goverment authorities?

l94 which other ingredients (e.g.)..maybe ingredients () and/or strategies

l118 reference

l298 which sustainable practices

Good luck in your future work!

Best regards.

Author Response

Dear Editor

Thank you very much for your response to our manuscript “Strategies to reduce salt content: PDO and PGI meat products case” reference submission ID foods-3142818 by Maria João Fraqueza and colleagues.

We are also grateful for comments from the reviewers. We accordingly followed their suggestions and revised the manuscript, trying to provide a due account of all the concerns of the reviewer. All changes and corrections in the revised version of the manuscript were marked with track changes. Please find below our comments on the points raised by reviewer.

We hope this new revised version meets now the requirements for being accepted for publication in Foods journal.

Yours sincerely,

Maria João Fraqueza

Manuscript ID: foods-3142818

Title: Strategies to reduce salt content: PDO and PGI meat products case

Status: Pending minor revision

Response to the reviewer comments:

Thank you for your comments and suggestions. We appreciate it. We tried to address all of them.

Reviewer: Thank you for the thorough responses you provided. In addition, I have a few small suggestions that I hope you will find helpful.

l63 which government authorities?

Reply: The authors acknowledge the reviewer remark. Government authorities of the Member States that provide updates on their national salt reduction efforts.

l94 which other ingredients (e.g.)..maybe ingredients () and/or strategies

Reply: The reviewer is absolutely right. Alternative ingredients and/or strategies were already introduced in page 2, lines 94-95.

l118 reference

Reply: The authors added the reference Bi, Y.; Liang, L.; Qiao, K.; Luo, J.; Liu, X.; Sun, B.; Zhang, Y. A comprehensive review of plant-derived salt substitutes: Classification, mechanism, and application. Food Res Int 2024, doi: https://doi.org/10.1016/j.foodres.2024.114880, which was already a listed reference (previously reference number 64 and now as number 24).

l298 which sustainable practices

Reply: Environmental (good resources use, minimum waste, circular economy, maintenance of biodiversity, good management of cultural heritage, life cycle assessment), economic (business support, job creation and maintenance with development of rural areas), and social sustainable practices including animal welfare as indicated in page 9, line 395.

Reviewer 2 Report

Comments and Suggestions for Authors

Because the authors have revised this manuscript well, reflecting the opinions of reviewers, it is believed that this manuscript can be published in this journal.

Author Response

(The authors gave the same response as above.)
